# ActiveScope: Actively Seeking and Correcting Perception for MLLMs

**Yajing Wang** [1 2]  **Chao Bi** [2]  **Junshu Sun** [2]  **Shufan Shen** [2]  **Zhaobo Qi** [3]  **Shuhui Wang** [2]  **Qingming Huang** [1 2]

## Abstract

Multimodal Large Language Models (MLLMs) have demonstrated impressive vision-language understanding, yet still struggle with fine-grained perception in high-resolution images. While existing training-free methods typically rely on attention-based localization or coarse-to-fine search, they are often misled by distractors and fail to locate multiple targets. Our investigation attributes these failures to *Contextual Dominance*, where salient distractors overwhelm target attention and cause inaccurate localization, and *Semantic Bias*, where global semantics cause the model to fixate on the most salient concept, resulting in incomplete localization in multi-object scenarios. Built on these insights, we propose ActiveScope, a training-free framework that enhances MLLMs by actively seeking and correcting perception. ActiveScope features two modules. The *Semantic Anchor Localization (SAL)* utilizes fine-grained semantic anchors to independently localize key targets, thereby mitigating semantic bias. The *Interference-Suppressed Refinement (ISR)* refines localization by suppressing attention on salient distractions to overcome contextual dominance. Extensive experiments on high-resolution image understanding benchmarks demonstrate that ActiveScope outperforms existing training-free methods (e.g., 96.34% accuracy on $V^*$ Bench), validating the superiority of the active search and self-correction paradigm. Our code is available at https://github.com/jasmine-ww/ActiveScope.

## 1. Introduction

Multimodal large language models have demonstrated exceptional capabilities in visual understanding and reasoning across diverse tasks, including image captioning, visual question answering, and cross-modal retrieval (Bai et al., 2023a; Chen et al., 2024b; Yang et al., 2025). However, previous works (Zhang et al., 2025a; Shen et al., 2025) reveal that these models still struggle with fine-grained visual perception, which requires understanding detailed attributes or performing spatial reasoning on small target regions within a single high-resolution image.

To enhance MLLMs' perception of fine-grained visual details, existing methods generally adopt a localize-then-zoom-in paradigm, which first localizes the key region and then zooms in to assist the model in generating answers. Training-based methods encourage models to predict key regions via supervised fine-tuning (SFT) (Shao et al., 2024) or reinforcement learning (RL) (Zheng et al., 2026), requiring significant training costs and risking catastrophic forgetting. In contrast, for training-free methods, some methods localize informative regions by leveraging attention maps during MLLM inference (Zhang et al., 2025a). Alternatively, some research adopts search-based or retrieval strategies that scan images in a coarse-to-fine manner (Shen et al., 2025; Li et al., 2025a). However, these approaches are inherently inflexible, as early localization mistakes propagate to later stages, and they struggle to support sequential localization of multiple targets in multi-object scenarios.

In this paper, we systematically analyze the causes of localization failures from two perspectives. We categorize localization failures into two primary dimensions: inaccurate localization and incomplete localization in multi-target scenarios, as in Figure 1 (a). The former refers to the model's inability to precisely identify a specific target, while the latter denotes a failure to exhaustively locate all relevant entities, which is essential for complex spatial reasoning.

Regarding the inaccuracy of localization, we analyzed two potential causes: *local ambiguity* and *contextual dominance*, with the latter emerging as the primary bottleneck. Specifically, local ambiguity occurs when the attention score of the true target is marginally surpassed by a local peak from a semantically similar object; in contrast, contextual dominance represents a more dominant failure where visually salient distractors overwhelm the entire attention landscape, resulting in the target being missed. Consequently, a potential treatment is to actively identify and mask these misleading

---

[1]University of Chinese Academy of Sciences [2]State Key Lab of AI Safety, Institute of Computing Technology, Chinese Academy of Sciences, Beijing, China. [3]Harbin Institute of Technology (Weihai). Correspondence to: Shuhui Wang <wangshuhui@ict.ac.cn>.

*Proceedings of the 43rd International Conference on Machine Learning*, Seoul, South Korea. PMLR 306, 2026. Copyright 2026 by the author(s).

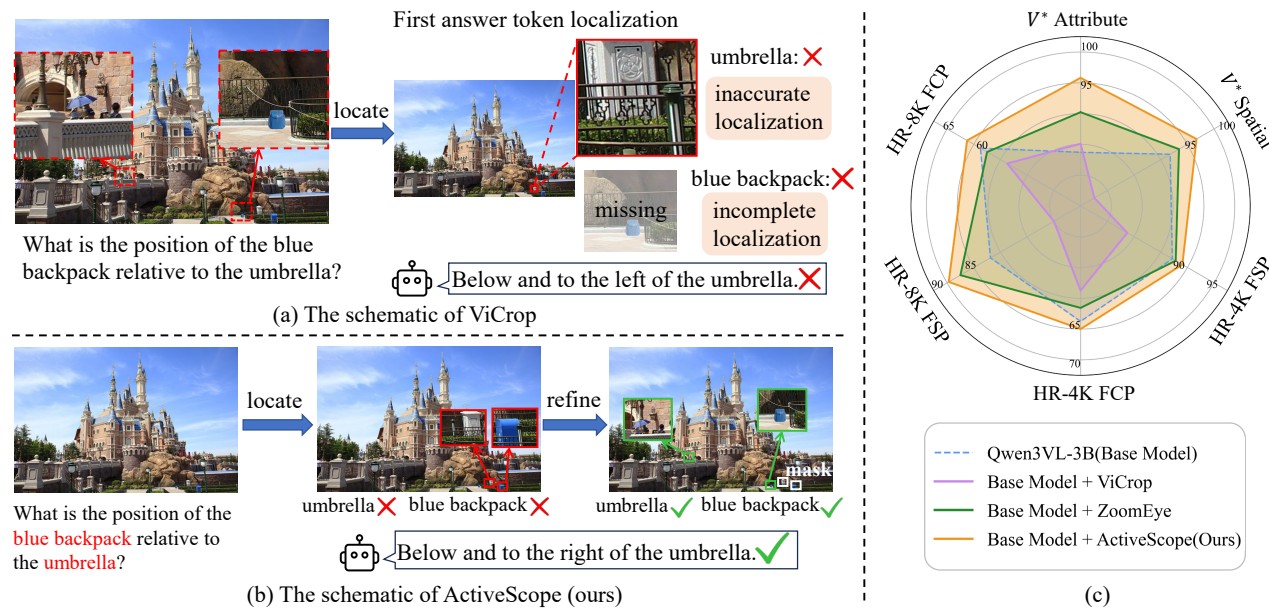

*Figure 1.* (a) Schematic of the previous training-free method, which struggles to locate hard targets and adapt to multi-target scenarios. (b) ActiveScope supports the localization of multiple targets and can refine the initial positioning to the correct regions. (c) ActiveScope outperforms other methods on multiple benchmarks.

areas to redistribute attention, thereby allowing the model to correct its focus.

In terms of incomplete localization in multi-object scenes, we identify a persistent *semantic bias*. In this case, attention derived from the first answer token aggregates the global semantics of the query and becomes dominated by the most salient concept, causing other relevant targets to be overlooked. Our study demonstrates that this issue can be resolved by anchoring the model's focus to specific descriptive keywords in the user's query to obtain corresponding attention maps. By treating each target's description as an independent semantic guide, the model can effectively decouple its attention and accurately ground multiple objects in complex environments.

Building on these insights, we propose ActiveScope, a training-free framework that enhances MLLMs by actively seeking and correcting perception. Unlike previous methods that rely on implicit global attention or brute-force search, ActiveScope explicitly guides the model to "look for" specific objects mentioned in the query through two core modules. First, the Semantic Anchor Localization (SAL) utilizes fine-grained semantics as anchors to independently localize key targets, thereby mitigating the issue of semantic bias. Subsequently, we introduce Interference-Suppressed Refinement (ISR) to ensure accurate localization and rectify erroneous predictions. By suppressing attention to dominant distracting areas, ISR guides the model to refine its focus

within a limited number of steps, thereby effectively alleviating contextual dominance. As illustrated in Figure 1 (b), ActiveScope identifies multiple targets and achieves precise localization through refinement, thereby enabling the model to correctly answer fine-grained visual questions.

By integrating ActiveScope into strong MLLMs such as Qwen3-VL and InternVL2.5, we evaluate its effectiveness on three challenging high-resolution benchmarks such as $V^*$ Bench, HR-Bench 4K, and HR-Bench 8K. Extensive experiments demonstrate that our method outperforms existing state-of-the-art methods, as in Figure 1 (c). For instance, on the $V^*$ Bench, ActiveScope achieves an accuracy of 96.34%, outperforming the best competing method by 3.15% and improving the base model by 6.81%, showcasing its superiority in fine-grained understanding of high-resolution images. Our contributions are summarized as follows:

- We systematically investigate the causes of localization failures, identifying contextual dominance in inaccurate localization and semantic bias in incomplete localization. Our analysis reveals that these bottlenecks can be effectively resolved by suppressing visually salient distractors and explicitly anchoring attention to facilitate precise and individual localization.

- We propose ActiveScope, a training-free framework composed of Semantic Anchor Localization (SAL) and Interference-Suppressed Refinement (ISR) to enable

active and self-corrective perception.

- ActiveScope achieves state-of-the-art performance on $V^*$ Bench and HR-Benches, validating that guiding MLLMs to actively "seek and correct" leads to a more effective perception strategy.

## 2. Related Work

### 2.1. Multimodal Large Language Models

Multimodal Large Language Models serve as the foundation for bridging vision and language modalities. The standard architecture typically consists of a pre-trained visual encoder (Dosovitskiy et al., 2021; Radford et al., 2021), a Large Language Model (LLM) (Touvron et al., 2023; Bai et al., 2023a), and a multimodal connector (Liu et al., 2023; Li et al., 2023). Early approaches generally process images at fixed, low resolutions (e.g., $224 \times 224$ or $336 \times 336$), which inevitably causes distortion and loss of fine-grained details. To mitigate this bottleneck, recent research has diverged into two main directions. One stream focuses on dynamic resolution adaptation, exemplified by models like Qwen2-VL (Bai et al., 2023a; Wang et al., 2024; Bai et al., 2025; Yang et al., 2025) and InternVL (Chen et al., 2024b;a; Zhu et al., 2025), which tokenize images into variable-length sequences to preserve native aspect ratios. The other stream integrates auxiliary high-resolution visual encoders (Li et al., 2025b; Luo et al., 2025; Ge et al., 2024; Liu et al., 2022), employing additional branches to explicitly capture high-frequency details without significantly increasing context length. Our work presents a training-free methodology that can be applied to existing MLLMs for enhanced perception. This approach provides orthogonal benefits, acting as a scalable complement to current high-resolution architectures.

### 2.2. Fine-grained Visual Understanding

To address the challenge of perceiving small objects or intricate details, existing methods generally fall into training-based and training-free paradigms. Training-based approaches aim to align models with fine-grained data distributions through Supervised Fine-Tuning (Shao et al., 2024; Wu & Xie, 2024) or incentivize "looking closer" via Reinforcement Learning (Zheng et al., 2026; Zhao et al., 2025; Fan et al., 2026). However, these parametric updates often incur high computational costs and risk catastrophic forgetting (Zhai et al., 2023; Chen et al., 2026), where the model's general capabilities degrade as it overfits to specific tasks.

On the other hand, training-free approaches require no additional parameter updates and can be flexibly extended to diverse architectures. These methods typically leverage a "locate-then-zoom-in" mechanism (Zhang et al., 2025a; Mao et al., 2025; Liu et al., 2025; Zhong et al., 2026), which first localizes the key region relevant to the query and subse-

quently crops or magnifies it to enhance fine-grained perception. A notable direction explores utilizing the model's inherent interpretability, as demonstrated in ViCrop (Zhang et al., 2025a), which leverages attention maps or gradients from the pre-trained model to identify and crop informative regions without external supervision. Alternatively, some methods employ various explicit search algorithms (Shen et al., 2025; Wang et al., 2025a; Li et al., 2025a; Zhou et al., 2025). ZoomEye (Shen et al., 2025) employs a tree-based search strategy to iteratively explore and identify target regions. Beyond search, recent works (Wang et al., 2025b; Gao et al., 2025; Yang & Zhang, 2025; Liu et al., 2024) have introduced retrieval mechanisms at varying granularities, from image patches to feature tokens, to supplement detailed information. However, these localization-dependent methods generally prioritize a single salient region, rendering them inadequate for multi-object queries. Furthermore, they are susceptible to error propagation, where initial localization inaccuracies compromise the final outcome. To bridge these gaps, our method facilitates MLLMs to actively seek latent multiple key regions and correct perception, enabling robust fine-grained understanding in complex scenarios.

## 3. Diagnosing Localization Bottlenecks

In this section, we conduct a systematic investigation to diagnose the fundamental reasons for MLLM failures in fine-grained visual perception. We study two distinct failure cases: (1) inaccurate localization, where the model fails to precisely locate a specific target due to visual or semantic distractions; and (2) incomplete localization in multi-target scenarios, where it fails to exhaustively identify all relevant targets. All preliminary experiments in this section are conducted using the Qwen3-VL model on the $V^*$ benchmark to provide empirical support for our analysis.

### 3.1. Analysis of Inaccurate Localization: Local Ambiguity vs. Contextual Dominance

Previous methods (Zhang et al., 2025a) typically localize salient regions by extracting the attention maps of the first predicted answer token and selecting the area with the highest activation score. However, we observe that this "one-shot" attention often highlights incorrect regions, leading to low precision. We hypothesize that these localization errors stem from two distinct mechanisms: local ambiguity and contextual dominance. Local ambiguity occurs when the true target is successfully activated but competes with semantically similar regions, causing its attention score to be marginally surpassed by a plausible distractor. Conversely, contextual dominance arises when the distractor or visually salient objects dominate the attention landscape, preventing the target from being effectively activated.

To validate these hypotheses and determine the key factor,

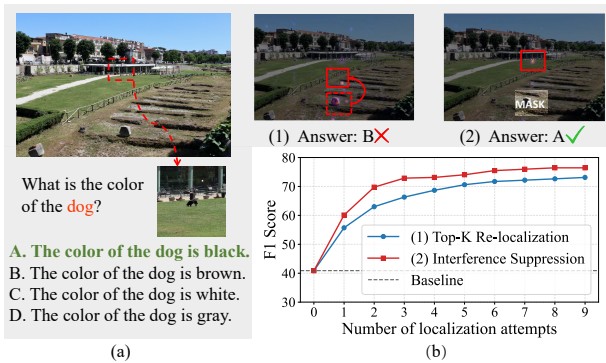

*Figure 2.* (a) Illustration of two strategies for iterative localization: (1) Top-K Re-localization and (2) Interference Suppression. In this example, the initial localization identifies a rock as a dog. With Top-K Re-localization, the second localization attempt still fails, whereas Interference Suppression masks the confusing region and successfully localizes the dog, leading to the correct answer. (b) Localization performance vs. Iteration count.

we designed two comparative strategies for iterative localization correction. To address local ambiguity, we employed Top-K Re-localization. Specifically, if the previous localization fails, we revise the prediction by selecting the region with the next highest attention score. To investigate contextual dominance, we implemented Interference Suppression, which masks the initially detected wrong area to perform a second inference pass, thereby forcing the model to redistribute attention and seek other potential candidates. The two strategies are shown in Figure 2 (a).

We conducted a $k$-step iterative experiment to analyze how localization performance evolves with increasing iterations, as shown in Figure 2 (b). The localization performance is measured by the F1 score based on the Intersection over Ground Truth (IoGT). Specifically, a prediction $R_{pred}$ is correct relative to a ground truth box $R_{gt}$ if the ratio of their intersection to the ground truth area exceeds 0.5.

Based on the trends observed in Figure 2 (b), three key observations can be made. First, both strategies demonstrate performance improvements, verifying the concurrent influence of both local ambiguity and contextual dominance. Second, Interference Suppression achieves a steeper ascent and higher ceiling, identifying contextual dominance as the primary bottleneck and suppression as the superior correction mechanism. Third, we observe that the most substantial gains are achieved within the first three iterations, indicating that the majority of correctable errors can be efficiently resolved with a small number of recursive steps.

### 3.2. Analysis of Incomplete Localization: Semantic Bias

In multi-object scenarios, the model must locate every target mentioned in the query to support relational reasoning, yet existing methods often fail to sequentially identify all

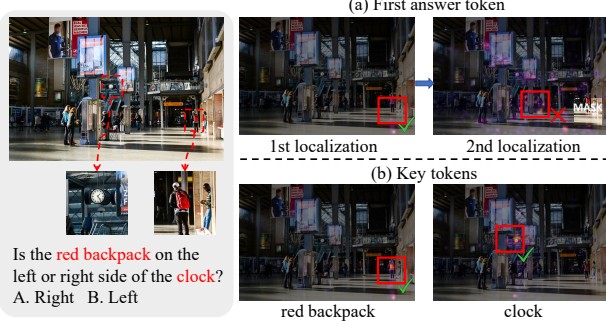

*Figure 3.* (a) iterative localization: localize with Interference Suppression strategy for two steps. (b) key token guidance: localize using attention maps from the key tokens corresponding to targets.

*Table 1.* Key region localization performance of different strategies. This table compares three strategies: baseline, iterative localization, and key token guidance. Performance is measured by precision, recall, and F1 score($\uparrow$). The best results are highlighted in bold.

|  | Spatial | | | Average | | |
|---|---|---|---|---|---|---|
|  | Rec | Pre | F1 | Rec | Pre | F1 |
| Baseline | 26.4 | 45.2 | 33.3 | 36.3 | 46.7 | 40.9 |
| iterative localization | 36.8 | 32.9 | 34.7 | 51.7 | 34.3 | 41.2 |
| key token guidance | **64.8** | **57.0** | **60.7** | **65.8** | **64.2** | **65.0** |

relevant targets. We observe that even when applying the Interference Suppression strategy, the model struggles to traverse distinct objects. As illustrated in Figure 3 (a), after the model correctly localizes a "red backpack", the corresponding region is masked to encourage exploring the next target "clock". However, instead of shifting focus to the "clock", the model erroneously localizes a "black backpack" in a nearby region. We attribute this failure to semantic bias. Specifically, since the attention map is derived from the first answer token, it implicitly aggregates the global semantics of the query. Therefore, the attention map tends to be monopolized by the most dominant semantic category, suppressing activations for other distinct targets.

To validate this, we compared three strategies: the baseline, iterative localization, and key token guidance. Specifically, the baseline localizes key area using the attention map of the first answer token. Building on this, the iterative localization strategy (Figure 3(a)) also utilizes the first answer token but employs two rounds of Interference Suppression to adapt to multi-object scenarios. In contrast, the key token guidance strategy (Figure 3(b)) derives attention maps from the tokens corresponding to the specific key objects, thereby enabling the localization of each individual target. We then evaluated the localization performance of the three strategies using precision, recall, and F1 score. As shown in Table 1, the quantitative results reveal that the key token guidance strategy achieves a substantial performance leap,

significantly outperforming the other strategy. While iterative localization improves recall and F1 score, it remains far inferior to the key token-based approach. This disparity confirms that the semantic bias in next-token prediction prevents comprehensive coverage of multiple relevant regions, highlighting the necessity of explicit semantic guidance for accurate multi-object grounding.

**Summary.** Based on the above analysis, we summarize two primary bottlenecks hindering fine-grained understanding. First, inaccurate localization is largely driven by contextual dominance, which inspires us to refine localizations by suppressing attention to interference regions. Second, incomplete localization in multi-object scenarios stems from semantic bias, which can be effectively mitigated by utilizing fine-grained semantic tokens as anchors to decouple and localize each target independently.

## 4. Method

### 4.1. Overview

Based on the analysis above, we propose ActiveScope, a training-free framework designed to enhance the fine-grained understanding of MLLMs. As illustrated in Figure 4, our method primarily consists of two key modules. First, to explicitly guide the model's attention toward the specific region of each relevant target, we introduce Semantic Anchor Localization. This component utilizes fine-grained keywords as semantic anchors to derive target-specific attention maps, thereby decoupling the positional information of multiple targets to mitigate semantic bias. Second, to resolve contextual dominance and address the limitations of one-shot localization, we propose Interference-Suppressed Refinement. This mechanism suppresses attention on incorrectly localized or confusing regions, thereby guiding the model to redistribute attention and refine its localization. Finally, the confirmed regions are composed into a focused visual prompt to assist in answering the question. In the following sections, we will introduce each module in detail.

### 4.2. Semantic Anchor Localization

As observed in section 3.2, relying on the first predicted answer token for critical region localization often suffers from semantic bias, a phenomenon where the aggregated semantics cause the model to repeatedly anchor to the same target rather than capturing all relevant regions simultaneously. In contrast, the attention maps of individual key target tokens contain more precise and targeted positional information. Consequently, the SAL module is designed to leverage these fine-grained semantic cues as anchors to accurately localize each individual target.

Given an input image and a query, we extract the image representation $X = \{x_1, x_2, ..., x_N\}$ and the text represen-

tation $T = \{t_1, t_2, ..., t_L\}$, respectively. We prompt the LLM with "If you want to answer the question, which objects' information do you need?" to extract a set of key semantic targets, denoted as $\mathcal{O} = \{O_1, O_2, ..., O_K\}$. For a specific object $O_k = \{t_{s_k}, ..., t_{e_k}\} \subset T$, where $s_k$ and $e_k$ denote the start and end indices of the object description, respectively, we extract the cross-attention map from the $l$-th decoder layer of the MLLM. First, the attention weight distribution of a single text token $t_i$ to all image patches in the layer $l$ is formulated as:

$$\text{A}(t_i, X)^{(l)} = \text{Softmax}\left( \frac{t_i^{(l)} \cdot (X^{(l)})^\top}{\sqrt{d_k}} \right) \qquad (1)$$

where $t_i^{(l)}$ and $X^{(l)}$ denote the hidden states of the text token and image tokens at the $l$-th layer, respectively, and $d_k$ is the dimension of the attention head. To obtain the attention map for the object $k$, we aggregate the attention scores of its constituent semantic tokens:

$$\overline{A}(O_k, X)^{(l)} = \frac{1}{|O_k|} \sum_{t_i \in O_k} A(t_i, X)^{(l)} \qquad (2)$$

To alleviate the inherent noise in raw attention maps and ensure focus on target-relevant regions, we employ a relative attention mechanism (Darcet et al., 2024; Zhang et al., 2025a). Specifically, we encode a generic instruction $q'$ (e.g., "Describe the image") into a sequence of text features $T' = \{t'_1, t'_2, ..., t'_{L'}\}$, and then perform model inference to obtain the attention map of the first generated answer token, denoted as $A(t'_{L'+1}, X)^{(l)}$. We normalize the target attention map $\overline{A}(O_k, X)^{(l)}$ via element-wise division by this reference attention map, yielding the relative attention:

$$A_{rel}(O_k, X)^{(l)} = \frac{\overline{A}(O_k, X)^{(l)}}{A(t'_{L'+1}, X)^{(l)}} \qquad (3)$$

Based on the refined attention map, we employ multi-scale sliding windows to extract bounding boxes for key regions, with implementation details provided in the Appendix.

### 4.3. Interference-Suppressed Refinement

The Semantic Anchor Localization module yields initial candidate regions, yet these preliminary results may still localize irrelevant or incorrect semantic areas. To address this, we introduce Interference-Suppressed Refinement (ISR), which employs a corrective cycle of Verification and Interference Suppression. Specifically, for a localized region, we first verify the presence of the corresponding target. If the target is absent, which implies a false area, we trigger Interference Suppression. This mechanism masks the incorrect

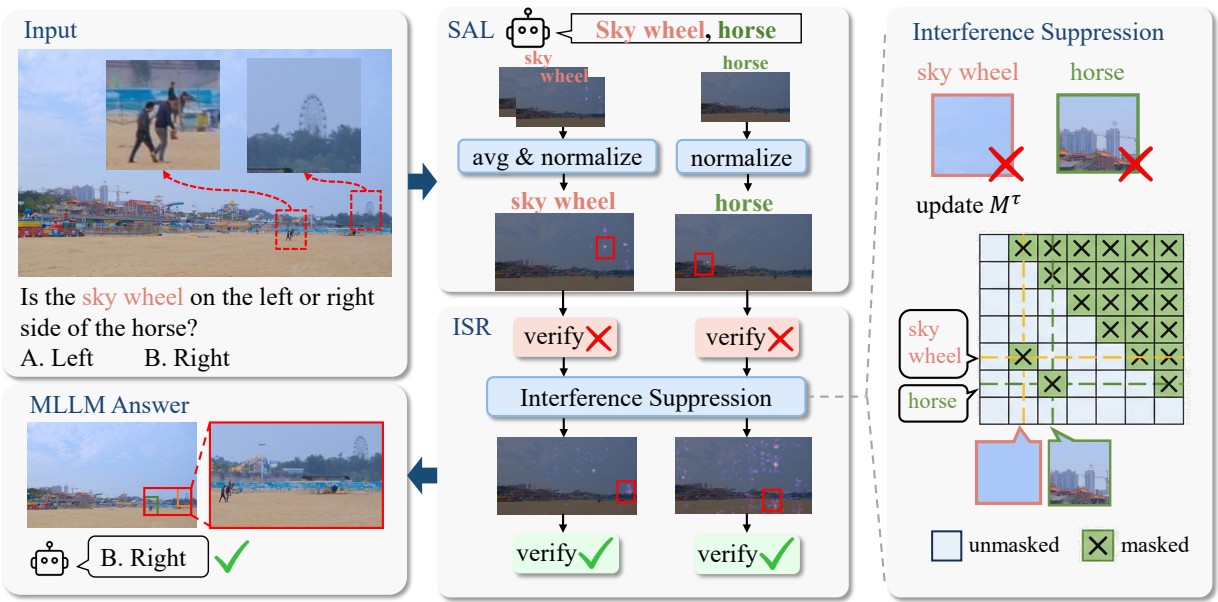

*Figure 4.* The framework of ActiveScope. SAL generates target-specific attention maps using semantic anchors, and ISR refines localization by suppressing interfering regions. The refined regions are assembled into a focused visual prompt for final prediction.

region, guiding the model to shift its attentional focus to other potential areas for re-localization. This process repeats for a maximum of $C$ cycles. In practice, we set $C \leq 3$ as the model often efficiently converges to the correct semantic area within a small number of iterations.

**Verification.** For a candidate box $b_i$ associated with target $O_i$, we crop the image to generate a local view $X_{crop}$. We then prompt the MLLM with a verification question (e.g., "Does $O_i$ exist in this image?") to validate the semantic alignment between the visual region and the text target.

**Interference Suppression.** If the verification fails, we suppress the attention from the current target to this confusing region to improve the next localization attempt. Specifically, we maintain a global attention mask matrix $M \in \mathbb{R}^{(N+L) \times (N+L)}$, where $N$ and $L$ denote the number of visual tokens and text tokens, respectively.

First, $M^{(0)}$ is initialized according to the standard multimodal sequence processing, where the mask ensures appropriate cross-modal and causal dependencies. When a target $O_k = \{t_{s_k}, ..., t_{e_k}\}$ is incorrectly localized to a set of visual tokens $X_{fail}$ inside the rejected box, we update the mask to explicitly block the attention between $O_k$ and $X_{fail}$. The mask for the current cycle $\tau$ is updated recursively based on the previous state $M^{(\tau-1)}$:

$$M_{j,h}^{(\tau)} = \begin{cases} -\infty, & \text{if } j \in [s_k, e_k] \text{ and } x_h \in X_{fail} \\ M_{j,h}^{(\tau-1)}, & \text{otherwise} \end{cases} \quad (4)$$

The updated mask $M^{(\tau)}$ is applied to all layers of the MLLM during the next inference for localization. By setting these specific attention scores to $-\infty$, the model is forced to ignore the confusing region and attend to the next most probable area in the subsequent localization step.

### 4.4. Composition and Final Inference

After the Interference-Suppressed Refinement, we obtain a set of verified bounding boxes. To provide the model with both global context and fine-grained details, we construct a composite input. We compute the union of all verified boxes to form a minimum enclosing rectangle, crop this region to create a "zoomed-in" view $X_{zoom}$. The final answer is generated by providing the MLLM with the original image $X$, the focused view $X_{zoom}$, and the original question $q$. This ensures the model utilizes the purified and verified visual information to answer fine-grained visual queries.

## 5. Experiments

### 5.1. Benchmarks

We evaluate our proposed method on four challenging high-resolution benchmarks: $V^*$ Bench (Wu & Xie, 2024), HR-Bench 4K, HR-Bench 8K (Wang et al., 2025a) and MME-RealWorld-Lite (Zhang et al., 2025b). With an average resolution of $2246 \times 1582$, $V^*$ Bench provides two sub-tasks: attribute recognition and spatial reasoning. HR-Bench 8K features an average resolution of 7680, and HR-Bench 4K is

*Table 2.* Performance comparison on high-resolution benchmarks. "Regular" in method denotes the direct evaluation of the base model. For $V^*$ Bench, we report accuracy for attribute recognition and spatial reasoning tasks. HR-Bench includes Fine-grained Single-instance Perception and Fine-grained Cross-instance Perception tasks. The best results are highlighted in **bold** and the second is underlined.

| Method | Model | $V^*$ Bench | | | HR-Bench 4K | | | HR-Bench 8K | | |
|---|---|---|---|---|---|---|---|---|---|---|
| | | Attr | Spatial | Avg | FSP | FCP | Avg | FSP | FCP | Avg |
| Regular | Qwen-VL-max | - | - | - | 65.00 | 52.00 | 58.50 | 54.00 | 51.00 | 52.50 |
| Regular | GPT4o | - | - | 66.00 | 70.00 | 48.00 | 59.00 | 62.00 | 49.00 | 55.50 |
| Regular | Qwen3VL 4B | 86.96 | 93.42 | 89.53 | 88.75 | 65.00 | 76.88 | 83.50 | 60.00 | 71.75 |
| ViCrop | Qwen3VL 4B | 86.96 | 92.11 | 89.01 | 83.25 | 56.75 | 70.00 | 79.75 | 58.00 | 68.88 |
| ICoT | Qwen3VL 4B | 76.85 | 75.00 | 76.11 | 62.00 | 46.00 | 54.00 | 66.00 | 44.00 | 55.00 |
| ZoomEye | Qwen3VL 4B | 92.17 | 94.74 | 93.19 | 89.25 | 63.25 | 76.25 | 88.00 | 59.00 | 73.50 |
| Ours | Qwen3VL 4B | **95.65** | **97.37** | **96.34** | **90.25** | **66.00** | **78.13** | **89.50** | **60.50** | **74.75** |
| Regular | InternVL2.5 4B | 68.70 | 68.42 | 68.59 | 77.50 | 53.75 | 65.63 | 63.00 | 49.50 | 56.25 |
| ViCrop | InternVL2.5 4B | 44.00 | 58.25 | 51.13 | 61.25 | 49.50 | 55.38 | 60.75 | 44.50 | 52.63 |
| ICoT | InternVL2.5 4B | 59.13 | 61.84 | 60.21 | 49.00 | 39.00 | 44.00 | 53.00 | 48.00 | 50.50 |
| ZoomEye | InternVL2.5 4B | **80.87** | 77.63 | 79.58 | 81.25 | 56.25 | 68.75 | 78.50 | 50.50 | 64.50 |
| Ours | InternVL2.5 4B | 80.00 | **81.58** | **80.63** | **82.00** | **56.50** | **69.25** | **82.00** | **53.25** | **67.63** |

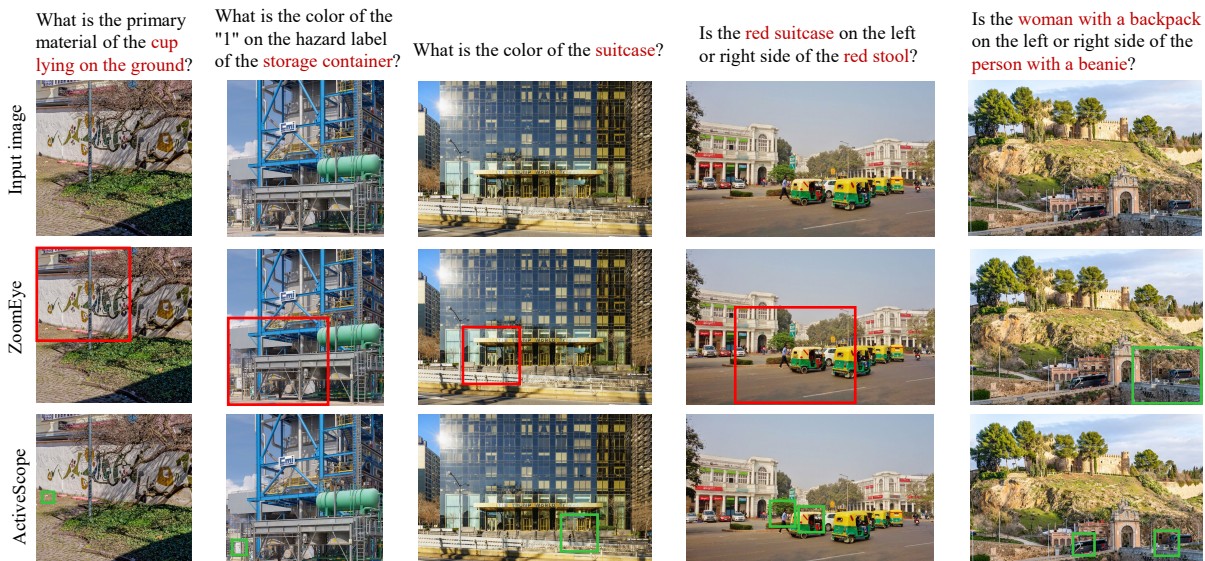

*Figure 5.* Qualitative comparison of localization results between ActiveScope and ZoomEye. Green and red boxes represent successful and failed localizations, respectively.

*Table 3.* Performance comparison on MME-RealWorld-Lite. The best score is highlighted in **bold**, and the second is underlined.

| Method | Model | MME-RW-Lite | | |
|---|---|---|---|---|
| | | Per | Rea | Avg |
| Regular | GPT-4o | **49.10** | 42.10 | 46.40 |
| Regular | Qwen3-VL 4B | 38.53 | 52.01 | 46.74 |
| ZoomEye | Qwen3-VL 4B | 39.73 | 55.60 | 49.40 |
| ActiveScope | Qwen3-VL 4B | 41.27 | **56.40** | **50.02** |

derived from it by cropping 8K images around relevant objects. Both benchmarks assess Fine-grained Single-instance Perception (FSP) and Fine-grained Cross-instance Perception (FCP). We also provide results for MME-RealWorld-Lite (Zhang et al., 2025b), which is designed for real-world applications with two subsets: perception and reasoning.

**5.2. Main Results**

To validate the effectiveness of our method, we compared it against two strong closed-source MLLMs, Qwen-VL-Max (Bai et al., 2023b) and GPT-4o (Achiam et al., 2023), as well as three representative training-free mechanisms: ViCrop (Zhang et al., 2025a), ICoT (Gao et al., 2025), and ZoomEye (Shen et al., 2025). Specifically, ViCrop utilizes the attention map of the first answer token to localize key positions, followed by cropping and zooming; ICoT selects important visual tokens to enhance local visual features, and ZoomEye iteratively searches for key regions in a coarse-

Table 4. Comparison of different localization strategies.

| Method | $V^*$ Bench | | |
|---|---|---|---|
| | Attr | Spatial | Avg |
| Qwen3-VL 4B(Base) | 86.96 | 93.42 | 89.53 |
| Base+predicted bbox | 87.18 | 83.96 | 84.94 |
| Base+next token bbox | 86.96 | 92.11 | 89.01 |
| Base+SAL (raw) | 88.70 | 93.20 | 91.91 |
| Base+SAL (full) | 93.03 | 94.76 | 93.73 |

Table 5. Ablation study for Interference-Suppressed Refinement.

| Method | $V^*$ Bench | | |
|---|---|---|---|
| | Attr | Spatial | Avg |
| w/o ISR | 93.03 | 94.76 | 93.73 |
| w/ ISR ($C$=2) | 94.35 | 96.06 | 95.01 |
| w/ ISR ($C$=3) | 95.65 | 97.37 | 96.34 |
| w/ ISR ($C$=4) | 95.65 | 97.37 | 96.34 |
| Top-K Re-localization | 93.91 | 96.05 | 94.76 |

to-fine manner. To ensure a fair comparison, we evaluated our method alongside other methods on two base models: Qwen3-VL 4B and InternVL2.5 4B.

As shown in Table 2, our method achieves superior performance across almost all metrics on both model architectures. Taking Qwen3-VL 4B as an example, our approach significantly outperforms the base model, achieving an average accuracy of 96.34% on $V^*$ Bench, which surpasses the performance of ZoomEye by 3.15%. We further evaluate ActiveScope on MME-RealWorld-Lite as shown in Table 3. Our method yields larger performance gains over the base model compared to the competing approach, and surpasses the strong closed-source model GPT-4o on most metrics. These results demonstrate that ActiveScope can localize critical regions more accurately, enhancing various base models' fine-grained perception, even outperforming tree-search-based mechanisms like ZoomEye.

### 5.3. Qualitative Analysis

To qualitatively demonstrate the effectiveness of our approach, we visualize the localization results of our method compared to the state-of-the-art ZoomEye in Figure 5. As observed, ZoomEye often produces imprecise or misaligned localizations in cases where objects are small or easily confused with surrounding regions. In contrast, our method demonstrates consistent precision in pinpointing target areas. In multi-target scenarios, specifically in the fourth and fifth examples, our approach successfully identifies all relevant entities simultaneously, whereas ZoomEye frequently fails to detect some of the targets.

### 5.4. Ablation Study

We conduct ablation studies on the $V^*$ Bench using the Qwen3-VL 4B model to validate the effectiveness of our core components: SAL and ISR.

In order to evaluate the Semantic Anchor Localization module, We compare the following strategies: (1) Base model, (2) Base + predicted bbox: utilizing bounding boxes predictions generated by the base model, (3) Base + next token bbox: using the bounding boxes derived from the attention

Table 6. Verification performance with different prompts. We compare three distinct prompt variations ($p_1$, $p_2$, and $p_3$) used for the "object exists" check. The table reports the verification accuracy, false positive rate, and false negative rate to demonstrate the robustness of the verification process.

| Prompt | Acc(%) | FPR (%) | FNR (%) |
|---|---|---|---|
| $p_1$ | 95.8 | 10.7 | 1.0 |
| $p_2$ | 95.5 | 9.5 | 2.9 |
| $p_3$ | 95.6 | 10.7 | 1.1 |

map of the first answer token, which is similar to ViCrop (Zhang et al., 2025a), (4) Base + SAL (raw): implementing SAL module but using unnormalized (raw) attention maps, and (5) Base + SAL (full): our proposed SAL with normalization. As shown in Table 4, the proposed full SAL achieves superior performance, validating the efficacy of the module and the necessity of the normalization step.

We investigate the performance gain brought by Interference-Suppressed Refinement(ISR), and examine the impact of its key components: the maximum cycle count $C$ and the hard attention masking strategy. As shown in Table 5, incorporating ISR leads to an improvement in localization accuracy. We first analyze the impact of the cycle count $C$. Notably, we observe that the performance scales with $C$ but quickly saturates at $C = 3$, suggesting that the model successfully converges to the optimal semantic regions within very few iterations. To isolate the contribution of hard attention masking, we fix $C = 3$ and compare our approach against the Top-$K$ re-localization strategy (detailed in Section 3.1). Compared to Top-$K$, which simply selects boxes with the next highest attention scores, our hard masking strategy achieves superior performance, verifying that explicit interference suppression is critical to the refinement gains.

We also quantitatively evaluated the reliability of the verification process. During verification, we query the MLLM to determine whether a specific target exists within the cropped bounding box image. As described in Section 3.1, a target is considered truly present within a predicted bounding box if its Intersection over Union (IoU) with the ground truth exceeds 0.5. As shown in Table 6, the model achieves a high

*Table 7.* Efficiency comparison between ZoomEye and Ours. We report total inference time on $V^*$ Bench, the average number of VQA calls, and the peak GPU memory usage.

| Method | Inference Time | VQA calls | GPU Memory |
|--------|----------------|-----------|------------|
| Regular | 14min | 1 | 36GB |
| ZoomEye | 27min | 33 | 36GB |
| Ours | 24min | 4.7 | 40GB |

verification accuracy ($> 95\%$) with low false positive and false negative rates, confirming that it rarely hallucinates during this stage. Furthermore, to assess the prompt sensitivity of this "object exists" check, we evaluate the verification performance under three distinct prompt variations: (i) P1 (Ours), *Does the target exist in this image? Answer Yes or No.*; (ii) P2, *Does this image contain the target? Answer Yes or No.*; and (iii) P3, *Is the target present in this cropped image region? Answer Yes or No.* The verification accuracy remains highly stable across these different queries, demonstrating that the correctness of our ISR verification process is robust and largely insensitive to prompt modifications.

### 5.5. Efficiency Analysis

We compare the inference efficiency of our method to the search-based method, ZoomEye. We measure the total inference time on $V^*$ Bench, the average number of VQA queries required to reach an answer, and the peak GPU memory usage. As shown in Table 7, ZoomEye suffers from prolonged latency and excessive VQA calls due to its costly tree-search mechanism. Conversely, while ActiveScope introduces a multi-stage pipeline, its additional overhead remains highly manageable. Specifically, localization merely requires computing text-vision cross-attention up to intermediate layers, avoiding full inference and generation cost. During verification, the input bounding box crops are significantly smaller than the original high-resolution image, keeping computational costs modest. Compared to ZoomEye, ActiveScope requires less inference time and achieves more substantial performance gains, with only an approximately 4GB increase in GPU memory. As a result, our method provides a superior balance of efficiency and precision.

## 6. Conclusion

In this paper, we diagnose two primary localization bottlenecks for fine-grained perception in MLLMs: (1) Contextual Dominance, where salient distractors overwhelm attention and cause inaccurate localization; and (2) Semantic Bias, where the first answer token aggregates biased global semantics, leading to incomplete localization. To address these limits, we propose ActiveScope, featuring Semantic Anchor Localization for multi-target identification by the guidance of fine-grained semantics and Interference-Suppressed Refinement to refine localization by suppressing visual interference. Superior performance across multiple benchmarks validates that our "seek and correct" strategy effectively enhances fine-grained perception in complex scenarios.

## Acknowledgements

This work was supported in part by the National Key R&D Program of China under Grant 2023YFC2508704, in part by the National Natural Science Foundation of China under grant numbers 62236008, 62441232 and 62306092, in part by the Natural Science Foundation of Beijing under grant number L251082, and in part by Shandong Provincial Natural Science Foundation under project ZR2025ZD01.

## Impact Statement

This paper presents work whose goal is to advance the field of Machine Learning. There are many potential societal consequences of our work, none of which we feel must be specifically highlighted here.

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

## A. Algorithm of ActiveScope

The pseudocode for the ActiveScope method is provided below.

---

**Algorithm 1** Complete Algorithm Workflow of ActiveScope

---

**Input** : Image $I$, Query $Q$, MLLM mask $\mathcal{M}$, Iteration Limit $C$
**Output** : Final response $Y_{response}$

```
/* Step 1:  Extract Semantic Anchors using extraction prompt p_extract      */
```
1   $S \leftarrow \mathcal{M}.\text{generate}(I, p_{extract})$
2   $S_{bbox} \leftarrow \emptyset$
3   Initialize Attention Mask $M$ with standard cross-modal settings
4   **for** $iter \leftarrow 1$ **to** $C$ **do**
5     **if** $S = \emptyset$ **then**
6       **break**
7     **end**
```
      /* Get attention maps for specific targets and global context       */
```
8     $Attn_{map} \leftarrow \mathcal{M}.\text{forward}(I, Q, M)$
9     $Attn_{global} \leftarrow \mathcal{M}.\text{forward}(I, p_{global}, M)$
10     $a_g \leftarrow Attn_{global}[\text{last\_token}]$
11     **for** $target \in S$ **do**
12       Indices $\leftarrow \text{GetTokenIndices}(target)$
13       $a_{target} \leftarrow \text{Mean}(Attn_{map}[\text{Indices}])$
14       $A_{rel} \leftarrow a_{target}/a_g$            `// Normalize to get Relative Attention`
15       $b \leftarrow \text{ExtractBBox}(A_{rel})$
16       $I_{crop} \leftarrow \text{Crop}(I, b)$
```
        /* Verification Step:  Ask MLLM if target exists in crop             */
```
17       $q_{ver} \leftarrow \text{Format}(p_{verify}, target)$
18       $is\_verified \leftarrow \mathcal{M}.\text{generate}(I_{crop}, q_{ver})$
19       **if** $is\_verified$ **then**
20         $S_{bbox}.\text{add}(\{target, b\})$
21         $S.\text{remove}(target)$
22       **end**
23       **else**
```
          /* Interference Suppression                                        */
```
24         $\text{Indices}_{vis} \leftarrow \text{GetVisualTokensInBox}(B)$
25         **for** $j \in Indices$ **and** $h \in Indices_{vis}$ **do**
26           $M[j, h] \leftarrow -\infty$        `// Block attention between target and fail region`
27         **end**
28       **end**
29     **end**
30   **end**
31   $b^* \leftarrow$ Union bounding-boxes in $S_{bbox}$
32   $I_{crop^*} \leftarrow \text{Crop}(I, b^*)$
33   $Y_{response} \leftarrow \mathcal{M}.\text{generate}([I, I_{crop^*}], Q)$
34   **return** $Y_{response}$

---

## B. Implementation Details

All experiments were conducted on a single NVIDIA A100 GPU. For the evaluation on V* Bench, we set the maximum refinement cycles to $C = 3$, while for HR-Bench, the limit is set to $C = 2$. Regarding model-specific configurations, attention maps are extracted from layer $l = 22$ for Qwen3-VL 4B and layer $l = 27$ for InternVL2.5 4B. Notably, since InternVL2.5 employs a dynamic resolution strategy that divides images into multiple tiles for separate encoding, we aggregate the attention maps from individual tiles, which are then spatially reassembled to reconstruct the complete attention

map corresponding to the original image resolution, facilitating subsequent localization.

## C. Detailed Procedure for Bounding Box Extraction

We transform the attention maps into discrete bounding boxes through a multi-scale sliding window strategy. Specifically, we define various window sizes ranging from $1\times$ to $2\times$ the base image resolution. We perform a dense sliding window operation (stride = 1) to locate the peak activation area for each scale. The final selection is based on the maximum sensitivity of the internal attention sum; we choose the scale where the difference between the window's total attention and the average of its adjacent positions is maximized. The resulting crop is then normalized to the model's input size and integrated into the inference pipeline.

## D. More Results on Additional Benchmarks

We provide additional results to further examine the effectiveness of ActiveScope under different visual understanding scenarios. We include the Remote Sensing and Monitoring categories from MME-RealWorld-Lite to assess performance in dense scenes. We additionally report results on OCR-heavy tasks, including TextVQA and DocVQA, and visual reasoning tasks, including GQA and VSR. As shown in Table 8, ActiveScope consistently improves upon the base model and achieves larger performance gains than the compared method, demonstrating the robustness and general applicability of our active localization and refinement strategy.

*Table 8.* Performance comparison on additional datasets.

| Method | Model | TextVQA | DocVQA | GQA | VSR | MME-RW-Lite | | |
| --- | --- | --- | --- | --- | --- | --- | --- | --- |
| | | | | | | P-Mt | P-RS | Avg |
| Regular | Qwen3-VL 4B | 85.56 | 93.44 | 72.34 | 78.14 | 32.29 | 42.67 | 46.74 |
| ZoomEye | Qwen3-VL 4B | 86.21 | 94.01 | 72.82 | 78.66 | 37.30 | 46.00 | 49.40 |
| Ours | Qwen3-VL 4B | 86.77 | 95.12 | 73.01 | 79.00 | 38.32 | 47.67 | 50.02 |

## E. Scaling to Larger and MoE Models

To demonstrate the scalability of our approach, we apply ActiveScope to a larger dense model (InternVL2.5-26B) and a Mixture-of-Experts model (Qwen3VL-30B-A3B). As shown in Table 9, our method maintains a clear superiority over both the standard base models and ZoomEye, confirming the robust effectiveness of our method across diverse model scales and architectures.

*Table 9.* Performance comparison on larger-scale and MoE models.

| Method | Model | $V^*$ | | | HR4K | | | HR8K | | |
| --- | --- | --- | --- | --- | --- | --- | --- | --- | --- | --- |
| | | Attr | Spat | Avg | FSP | FCP | Avg | FSP | FCP | Avg |
| Regular | | 73.91 | 72.37 | 73.30 | **83.50** | 60.75 | 72.13 | 73.25 | 61.50 | 67.38 |
| ZoomEye | InternVL2.5-26B | 74.78 | 73.68 | 74.35 | **83.50** | 63.00 | 73.25 | 75.75 | 63.00 | 69.38 |
| Ours | | **80.00** | **75.00** | **78.01** | 81.50 | **68.00** | **74.75** | **79.25** | **63.25** | **71.25** |
| Regular | | 86.73 | 86.49 | 86.63 | 88.25 | 63.50 | 75.88 | 86.75 | 59.25 | 73.00 |
| ZoomEye | Qwen3VL-30B-A3B(MoE) | **90.43** | 85.53 | 88.48 | 89.25 | 64.25 | 76.75 | 88.25 | 59.50 | 73.88 |
| Ours | | 89.38 | **90.54** | **89.84** | **90.00** | **65.25** | **77.50** | **89.75** | **60.25** | **75.00** |

## F. Selection and Ablation of Attention Layers

In this section, we detail the attention layer selection strategy and present corresponding ablation studies to validate our choices. Existing research reveals that attention maps extracted from deeper layers generally exhibit more precise spatial localization. Guided by this, we randomly sampled 50 instances and compared the bounding box F1 scores of attention

maps from the 18th layer (the middle layer) to the final layer. As summarized in Table 10, peak performance is achieved at layer 22 for Qwen3-VL 4B and layer 27 for InternVL2.5 4B. Furthermore, ActiveScope demonstrates strong robustness across adjacent layers. Even when utilizing suboptimal layers, it consistently outperforms both the baselines and ViCrop, confirming that our framework is broadly applicable and insensitive to hyperparameter tuning.

*Table 10.* Ablation study on attention layer selection on the $V^*$ Bench.

| Method | Model | Layer | $V^*$-Attr | $V^*$-Spat | $V^*$-Avg |
|---|---|---|---|---|---|
| Regular | Qwen3VL 4B | - | 86.96 | 93.42 | 89.53 |
| ViCrop | Qwen3VL 4B | 22 | 86.96 | 92.11 | 89.01 |
| | | 21 | **96.52** | 93.43 | 95.29 |
| ActiveScope | Qwen3VL 4B | 22 | 95.65 | **97.37** | **96.34** |
| | | 23 | **96.52** | 94.74 | 95.81 |
| | | 24 | 93.91 | 93.43 | 93.72 |
| Regular | InternVL2.5 4B | - | 68.70 | 68.42 | 68.59 |
| ViCrop | InternVL2.5 4B | 27 | 44.00 | 58.25 | 51.13 |
| | | 26 | 74.80 | **82.89** | 78.01 |
| ActiveScope | InternVL2.5 4B | 27 | **80.00** | 81.58 | **80.63** |
| | | 28 | 76.52 | 81.58 | 78.53 |
| | | 29 | 78.26 | 80.26 | 79.06 |

## G. Distinctions Between ActiveScope and FOCUS

In this section, we compare the Interference-Suppressed Refinement (ISR) module in our approach with the verification process in FOCUS (Zhong et al., 2026). Although both methods utilize an iterative verification strategy to locate target objects, they differ fundamentally in their core mechanisms and computational efficiency.

**Mechanism: Dynamic Intervention vs. Static Ranking.** FOCUS employs verification as a static scorer to rank a pool of candidate Regions of Interest (ROIs). Consequently, if a salient distractor overwhelms the initial attention map, the true target may never be proposed, causing the static ranking to fail. In contrast, ISR treats verification failures as negative feedback. By dynamically suppressing attention on salient distractions, ISR breaks this "contextual dominance" and forces the model to redistribute its attention to effectively discover targets.

**Efficiency and Convergence.** FOCUS relies on evaluating numerous candidates, often requiring ten or more forward passes to achieve high recall, which introduces substantial computational overhead. Instead of passively ranking guesses, ISR actively seeks and refines localization. As demonstrated in Table 5 of the main text, our approach is significantly more efficient, effectively reaching its performance upper bound within merely three refinement cycles ($C \leq 3$).

## H. Applicability to Closed-Source Models

ActiveScope primarily leverages the internal representations of open-source MLLMs, aligning with existing perception-enhancing methods such as ViCrop and ICoT. Nevertheless, we also validate its applicability to closed-source models through an approximate implementation via iterative Question-Answering. Specifically, since internal attention maps are inaccessible, we replace them with explicit bounding box predictions generated by the model. We then prompt the model with its historical erroneous predictions to iteratively refine its localization.

*Table 11.* Comparison between Regular and ActiveScope using GPT-4o

| | | V*-Attr | V*-Spat | V*-Avg |
|---|---|---|---|---|
| Regular | GPT-4o | 67.83 | 63.16 | 65.97 |
| ActiveScope | GPT-4o | **68.26** | **64.47** | **66.75** |

As shown in Table 11, this approach yields consistent improvements on GPT-4o, empirically validating that the underlying

principles of ActiveScope generalize effectively even to closed-source architectures.

## I. Limitation

Despite the demonstrated efficacy of ActiveScope in addressing fine-grained visual perception challenges, our current framework presents certain limitations. Primarily, the reliance on rectangular bounding boxes to delineate important regions imposes a geometric constraint; this approach may prove sub-optimal for targets with highly irregular or non-convex shapes, where a rigid rectangular crop inevitably includes irrelevant background noise. In addition, while ActiveScope significantly improves performance on detailed visual understanding benchmarks, it may not yet be universally applicable to every distinct image category or specialized downstream task.

## J. Qualitative Examples of the ActiveScope Localization Process

We provide qualitative visualizations of the localization process on several examples. As shown in Figures 6 and 7, the model localizes each target object involved in the query and, through a self-correction process, ultimately converges to the correct regions, enabling accurate question answering.

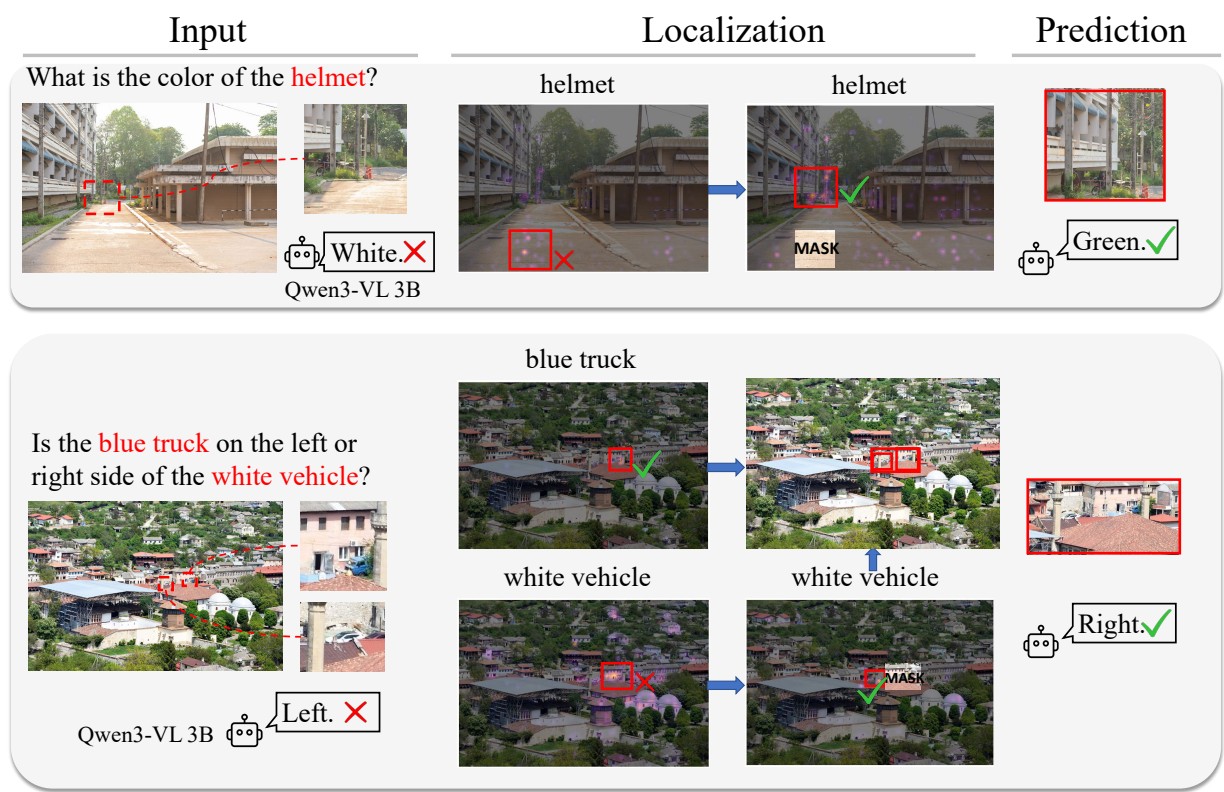

*Figure 6.* Visualization of iterative localization and self-correction across multiple target objects.

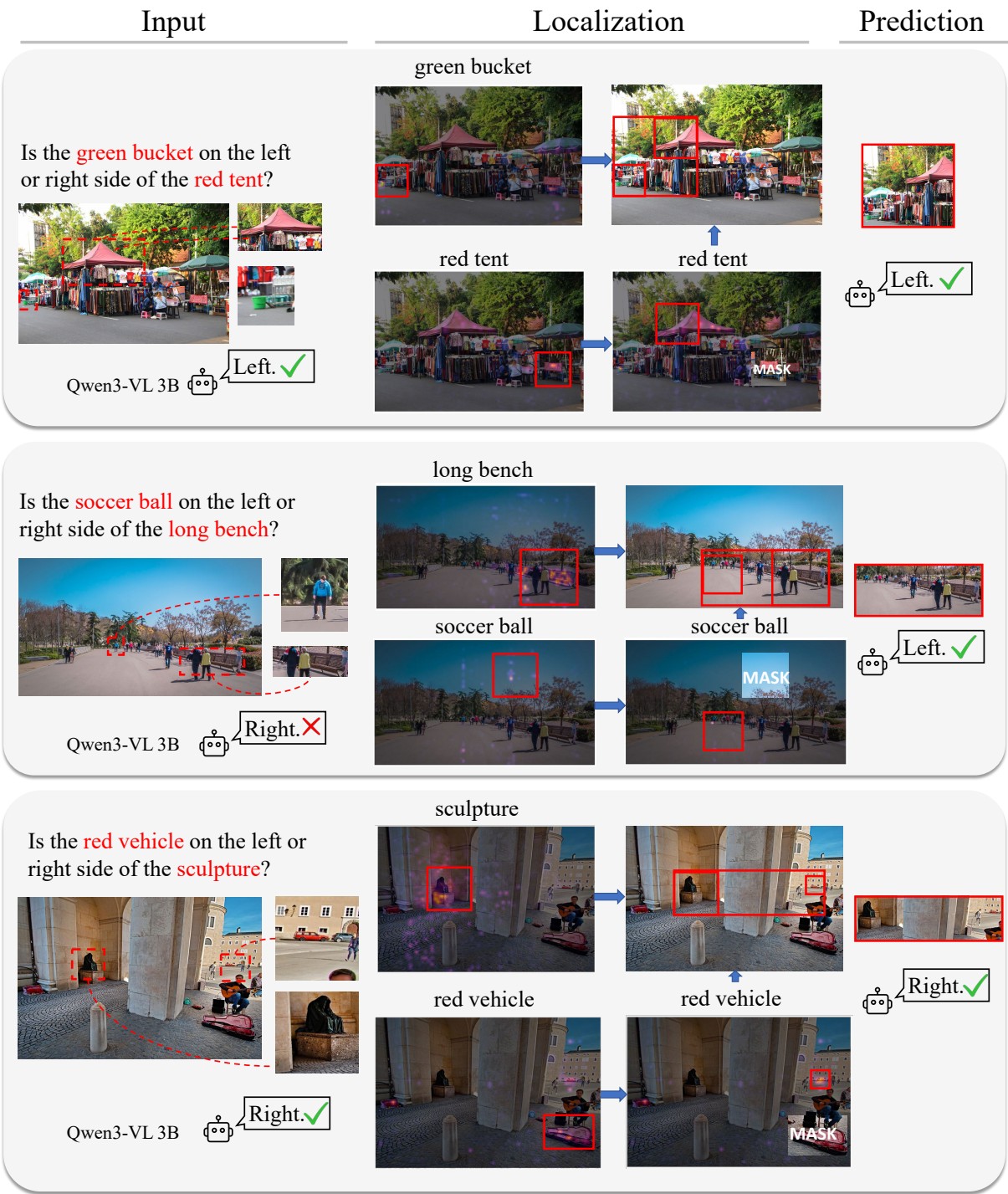

*Figure 7.* Visualization of iterative localization and self-correction across multiple target objects.

