# OpenReview forum: "ActiveScope: Actively Seeking and Correcting Perception for MLLMs"
_ICML.cc/2026/Conference — ICML 2026 regular_

### Official Review · Reviewer_MvN1 · 2026-03-06

**Soundness:** 3
**Presentation:** 3
**Significance:** 3
**Originality:** 2
**Overall Recommendation:** 4
**Confidence:** 4

**Summary:**

This paper addresses the issue of insufficient fine-grained perception ability of  MLLMs in high-resolution images, revealing two main causes for failed localization: Contextual Dominance and Semantic Bias. Consequently, the paper proposes ActiveScope, which comprises two core modules: Semantic Anchor Localization (SAL) to locate targets and Interference-Suppressed Refinement (ISR) to iteratively refine localization by suppressing distractors. The final target area is cropped and fed into the MLLM along with the original image. On high-resolution benchmarks such as V* Bench (96.34%) and HR-Bench 4K/8K, it outperforms existing training-free methods (such as ZoomEye).

**Compliance With Llm Reviewing Policy:**

Affirmed.

**Final Justification:**

This rebuttal addresses most of my concerns, and the authors’ responses to my follow-up question are inspiring. I have raised the soundness score and final rating.

Also, I have read the reviews from other reviewers and responses. From my point of view, the authors also addresses most of other reviews' concerns.

**Key Questions For Authors:**

In addition to the aforementioned weaknesses, there are also the following questions:
1. The ISR component seems very similar to the verification in [1]. Please elaborate on the differences between the two.
2. ViCrop did not use Qwen3VL or InternVL2.5.  How were the layers selected during the reproduction of ViCrop?
3. ViCrop provides three methods: pure attention, gradient-weighted attention, and relative attention. Which method was used for comparison in this paper?

[1] FOCUS: Internal MLLM Representations for Efficient Fine-Grained Visual Question Answering. NeurIPS 2025.

**Limitations:**

yes

**Strengths And Weaknesses:**

Strengths:
1. The paper devotes a whole section to discussing the reasons for the failure of MLLMs in the context of high-resolution image question-answering, and preliminarily quantifies the impact of various factors.
2. The proposed method aligns with the analysis in Section 3, ensuring good consistency in the presentation.
3. The experimental design is relatively comprehensive, with ablation studies conducted on the two main components proposed. From the results, it can be seen that compared to ZoomEye, the number of VQA calls has decreased from 33 to 4.7.

Weaknesses:
1. The paper only states in the appendix that Qwen3-VL 3B uses layer 22, and InternVL2.5 4B uses layer 27, but there is no layer sensitivity study to explain why these layers were chosen and how significant this choice would impact on performance.
2. The triggering of ISR relies on asking MLLM "Does Oi exist in this image?" to determine whether the localization is correct. However, the yes/no judgment of MLLM on cropped images may itself have hallucinations, which is not discussed in the main text or limitations.
3. The paper focuses on the visual localization problem in high-resolution image understanding scenarios. All the evaluation sets used, V*Bench and HR-Bench, contain a relatively small amount of data (totaling less than 1800 entries). Therefore, it could be insufficient to assess whether the method will consistently improve performance on general high-resolution tasks.

---

> ### Author Rebuttal · Authors · 2026-03-30
>
> We sincerely thank you for your constructive feedback, and we address your comments point-by-point below:
>
> **W1: Layer Selection and Sensitivity:**
>
> Layer selection depends primarily on the inherent architectural differences between models. To ensure a fair comparison, we used layer 22 for Qwen3-VL 3B, directly aligning with the established ViCrop configuration. For InternVL2.5 4B, layer 27 was selected based on empirical evaluations on a small validation set.
>
> To demonstrate our method's robustness, we conducted a layer sensitivity analysis on the V* benchmark:
>
> | Model | Method | Layer| V*-Attr | V*-Spat| V*-Avg|
> |-----|----|----|-----|----|------|
> | Qwen3VL 3B | Regular | - | 86.96 | 93.42| 89.53|
> | | ViCrop |22| 86.96| 92.11| 89.01|
> |  | ActiveScope |21| **96.52** | 93.43| 95.29|
> |  | ActiveScope| 22| 95.65| **97.37** | **96.34** |
> |  | ActiveScope | 23| **96.52** | 94.74| 95.81|
> |  |ActiveScope| 24| 93.91| 93.43| 93.72|
> |  InternVL2.5 4B | Regular | - | 68.70 | 68.42| 68.59 |
> | | ViCrop | 27 | 44.00| 58.25| 51.13 |
> | | ActiveScope | 26| 74.80 | **82.89** | 78.01 |
> | | ActiveScope| 27| **80.00** | 81.58 | **80.63** |
> | | ActiveScope| 28 | 76.52 | 81.58 | 78.53 |
> | |ActiveScope | 29 | 78.26 |80.26 | 79.06 |
>
> The results indicate that while layers 22 and 27 yield optimal performance, ActiveScope maintains strong robustness across adjacent layers. Performance drops only slightly at suboptimal layers and consistently outperforms both the baseline and ViCrop.  We will add this ablation study to the revised manuscript.
>
> **W2: ISR Verification**
>
> To address concerns regarding potential MLLM hallucinations during ISR, we quantitatively evaluated verification accuracy. A predicted bounding box is considered correct if its Intersection over Union (IoU) with the ground truth exceeds 0.5.
>
> |  | verify_acc(%) | verify_fpr(%)  | verify_fnr(%)  |
> |--------|-----|------|-------|
> |  ActiveScope    | 95.8 | 10.7 | 1.0  |
>
> As shown above, the MLLM achieves high verification accuracy (>95%) with low false positive and false negative rates across all variants. This confirms that the model rarely hallucinates during cropped region verification.
>
> **W3: Generalization on Broad Tasks**
>
> To validate ActiveScope's generalizability beyond V*Bench and HR-Bench, we extended our evaluation to diverse general visual tasks. This includes dense scenarios (MME-RW-Lite Remote Sensing/Monitoring), OCR (TextVQA, DocVQA), and visual reasoning (GQA, VSR):
>
> | Mehod   |   Model    | TextVQA   |  DocVQA   | GQA       | VSR       | MME-RW-Lite Mt | MME-RW-Lite RS | MME-RW-Lite Avg |
> | ------- | :--------: | --------- | :-------: | --------- | --------- | :------------: | -------------- | --------------- |
> | Regular | Qwen3VL 3B | 85.56     |   93.44   | 72.34     | 78.14     |     32.29      | 42.67          | 46.74           |
> | ZoomEye | Qwen3VL 3B | 86.21     |   94.01   | 72.82     | 78.66     |     37.30      | 46.00          | 49.40           |
> | Ours    | Qwen3VL 3B | **86.77** | **95.12** | **73.01** | **79.00** |   **38.32**    | **47.67**      | **50.02**       |
>
> These results demonstrate that ActiveScope consistently improves performance across varied domains compared to the regular baseline and ZoomEye, proving its effectiveness in broader high-resolution and complex visual scenarios.
>
> **Q1: Differences between ISR and FOCUS's verification**
>
> While both ISR and the verification in FOCUS utilize a verification process to iteratively locate the target object, they differ fundamentally:
>
> **1. Mechanism: Dynamic Intervention vs. Static Ranking.**
>
> * **FOCUS** uses verification as a static scorer to rank a pool of candidate ROIs. If a salient distractor overwhelms the initial attention map, the true target may not be proposed, causing the static ranking to fail.
> * **ISR** treats verification failures as negative feedback and dynamically refines the localization by suppressing attention on salient distractions. This mechanism breaks "contextual dominance" and forces the model to redistribute attention to effectively discover targets.
>
> **2. Efficiency and Convergence.**
>
> * **FOCUS** relies on evaluating numerous candidates, often requiring up to or over 10 forward passes to achieve high recall, resulting in higher computational overhead.
> * **ISR** actively seeks and refines localization rather than passively ranking guesses, so it is more efficient. As shown in Table 5 in our paper, ISR effectively reaches its performance upper bound within just 3 refinement cycles ($C \le 3$).
>
> **Q2: ViCrop Layer Selection**
>
> For a strict and fair comparison, we reproduced ViCrop using the same layers applied to ActiveScope: layer 22 for Qwen3-VL 3B and layer 27 for InternVL2.5 4B.
>
> **Q3: ViCrop Method**
>
> During our reproduction of ViCrop, we utilized the "relative attention" method. This variant is generally more effective and also leverages attention maps for localization, ensuring the most equitable comparison.

---

> > ### Author Rebuttal · Reviewer_MvN1 · 2026-04-01
> >
> > This rebuttal addresses most of my concerns. I would very much like to raise my rating.
> >
> > Also, I have read the reviews from other reviewers and responses. From my point of view, the authors also addresses most of other reviews' concerns.
> >
> > I have one more follow-up question for the authors:
> >
> > Follow-up Question:
> > When it comes to deployment and real-world application, searching for optimal layer can be very costly and makes it difficult adapting to other tasks. Are there any preliminary ideas for a solution?

---

> > > ### Author Response · Authors · 2026-04-03
> > >
> > > We sincerely thank the reviewer for the positive feedback and this insightful follow-up question.
> > >
> > > For the concern about deployment and adaptation costs, we have observed that the optimal layer is inherently associated with the model's architecture rather than specific tasks. During our evaluations, we kept the selected layer constant across all tested datasets, and it consistently demonstrated strong and reliable localization performance. Therefore, the optimal layer only needs to be determined once per model, eliminating the need for repeated searches when adapting to new downstream tasks.
> > >
> > > We will also describe the specific strategy for layer selection. Existing research reveals that attention maps extracted from deeper layers generally exhibit more precise spatial localization. Based on this finding, we evaluate the bounding box F1 scores on 50 randomly sampled instances, checking from the middle layer (e.g., the 18th layer) to the final layer. This lightweight process easily pinpointed the optimal layers with very small computational cost.
> > >
> > > Therefore, adapting the model to different downstream tasks does not require repeated layer searching, and the low computational cost of the initial layer selection is acceptable.

---

### Official Review · Reviewer_K3No · 2026-03-07

**Soundness:** 3
**Presentation:** 3
**Significance:** 3
**Originality:** 3
**Overall Recommendation:** 5
**Confidence:** 3

**Summary:**

This paper presents a novel "Search-and-Refine" strategy designed to address the challenges of multi-object recognition. By leveraging Semantic Anchor Localization (SAL), the proposed method effectively handles object identification, while Interference-Suppressed Refinement (ISR) facilitates precise localization polishing. Experimental results demonstrate that this approach achieves state-of-the-art performance across multiple benchmarks.

**Compliance With Llm Reviewing Policy:**

Affirmed.

**Final Justification:**

My concerns have been addressed for the most part, and I maintain my positive assessment of this paper.

**Key Questions For Authors:**

I do not have any specific questions right now.

**Limitations:**

yes

**Strengths And Weaknesses:**

**Strengths**
1. This paper introduces a novel multimoda MLLM perception strategy that integrates searching and refinement mechanisms to achieve precise multi-object localization.

2. The experimental results are solid, showing strong performance across several key benchmarks compared to existing methods.

3. This paper is well-organized with clear logic, and the visualizations help in understanding the proposed method.

**Weaknesses**
1. The paper does not explain the choice of layers for extracting attention maps. Specifically, why were layer 22 for Qwen3-VL and layer 27 for InternVL2.5 selected? It would be helpful if the authors could provide some intuition or ablation studies regarding these choices.

The paper is well-executed, and I do not have other concerns at this stage. I look forward to reading the other reviewers' comments before making a final recommendation.

---

> ### Author Rebuttal · Authors · 2026-03-29
>
> We appreciate the reviewer's positive evaluation such as novel Search-and-Refine strategy, solid experimental results, and clear presentation. We will address your concerns below and will incorporate these clarifications and additional analyses into the revised manuscript.
>
> **Response to Weakness 1: Intuition and Ablation Study on Layer Selection**
>
> We will describe our strategy for layer selection and present an ablation study evaluating these choices.
>
> Existing research reveals that attention maps extracted from deeper layers generally exhibit more precise spatial localization. Guided by this, we randomly sampled 50 instances and compared the bounding box F1 scores of attention maps from the 18th layer (the middle layer) to the final layer. Based on these evaluations, layer 22 for Qwen3-VL 3B and layer 27 for InternVL2.5 4B yielded the most accurate localization results.
>
> For deeper investigation, we conducted an ablation study to analyze the sensitivity of ActiveScope to the choice of attention layers. The results on the V* Bench are summarized below:
>
> | Model | Method | Layer| V*-Attr | V*-Spat| V*-Avg|
> |-----|----|----|-----|----|------|
> | Qwen3VL 3B | Regular | - | 86.96 | 93.42| 89.53|
> | | ViCrop |22| 86.96| 92.11| 89.01|
> |  | ActiveScope |21| **96.52** | 93.43| 95.29|
> |  | ActiveScope| 22| 95.65| **97.37** | **96.34** |
> |  | ActiveScope | 23| **96.52** | 94.74| 95.81|
> |  |ActiveScope| 24| 93.91| 93.43| 93.72|
> |  InternVL2.5 4B | Regular | - | 68.70 | 68.42| 68.59 |
> | | ViCrop | 27 | 44.00| 58.25| 51.13 |
> | | ActiveScope | 26| 74.80 | **82.89** | 78.01 |
> | | ActiveScope| 27| **80.00** | 81.58 | **80.63** |
> | | ActiveScope| 28 | 76.52 | 81.58 | 78.53 |
> | |ActiveScope | 29 | 78.26 |80.26 | 79.06 |
>
> As shown in the table, the peak performance naturally aligns with our selected optimal layers. ActiveScope exhibits strong robustness across adjacent layers. The performance degrades slightly at suboptimal layers and consistently demonstrates improvements over both the baselines and the compared method ViCrop. This confirms that our approach does not rely on highly sensitive hyperparameters, and it can be treated as a broadly applicable solution.

---

> > ### Author Rebuttal · Reviewer_K3No · 2026-04-03
> >
> > I'd like to thank the authors for their rebuttal. My concerns have been addressed for the most part, and I maintain my positive assessment of this paper.

---

> > > ### Author Response · Authors · 2026-04-03
> > >
> > > We sincerely thank the reviewer for the constructive feedback. We are glad that our rebuttal has addressed your concerns, and we deeply appreciate your positive assessment.

---

### Official Review · Reviewer_xawK · 2026-03-10

**Soundness:** 3
**Presentation:** 3
**Significance:** 2
**Originality:** 2
**Overall Recommendation:** 4
**Confidence:** 4

**Summary:**

This paper analyzes fine-grained perception failures of MLLMs on high-resolution images, attributing them to distractor-dominated localization and incomplete localization for multi-object queries. It proposes ActiveScope, a training-free pipeline with Semantic Anchor Localization (SAL) using anchor-token cross-attention with relative normalization, and Interference-Suppressed Refinement (ISR) that iteratively verifies crops and applies hard attention masking to rejected regions before re-localizing.

**Compliance With Llm Reviewing Policy:**

Affirmed.

**Final Justification:**

I am glad that the authors have addressed my main concerns. I appreciate the additional clarifications and look forward to the release of the open source code. Therefore, I raise my overall recommendation from 3 to 4.

**Key Questions For Authors:**

1. If hard attention masking is removed/replaced (keeping the same number of refinement attempts), how much performance remains?
2. What are verification FP/FN rates and prompt sensitivity for the “object exists” check?
3. How robust is semantic anchor extraction to missing anchors?
4. How sensitive are results to the chosen attention layer and extraction parameters? and do you plan to release code for reproducibility?

**Limitations:**

No (technical limitations only; societal impact not discussed).

**Strengths And Weaknesses:**

Strengths

S1. Clear motivation and a coherent diagnosis-to-method storyline.
S2. SAL/ISR ablations indicate both help.

Weaknesses

W1. The causal contribution of hard masking is not well isolated (gains may partly come from extra refinement attempts).
W2. Critical dependencies (anchor extraction and verification) lack robustness reporting (FP/FN, prompt sensitivity), and implementation details for box-to-token mask injection are insufficient for reproducibility.

---

> ### Author Rebuttal · Authors · 2026-03-29
>
> We appreciate the reviewers’ constructive comments. We will address your concerns below and will incorporate these clarifications into the revised manuscript:
>
> **W1 \& Q1: Causal contribution of hard masking**
>
> To isolate the contribution of our hard attention masking (Interference Suppression) from the refinement process, we conducted an ablation study comparing three approaches: (1) Without Refinement: removing the ISR module to use the initial localization, (2) Top-K Re-localization: revising failed localization using the bounding box with the next highest attention score (as discussed in Section 3.1), and (3) Interference Suppression: our hard masking approach. Both refinement methods restrict the maximum cycle count to $C=3$. As shown in the table below, both refinement schemes lead to improvement. Our Interference Suppression strategy achieves the best performance. The results confirm that the gains are attributed to both the Interference Suppression and the refinement.
>
> | Method|V*-Attr|V*-Spat |V*-Avg|
> | ----- | ---- | --- | ---- |
> | w/o Refinement | 93.03| 94.76 | 93.73|
> | Top-K Re-localization| 93.91|96.05|94.76|
> | Interference Suppression (Ours)| **95.65**|**97.37**| **96.34**|
>
> **W2 \& Q2: Accuracy of verification and prompt sensitivity**
>
> To evaluate the accuracy of the verification process, we adopt the following criterion, i.e., we consider the target to truely present in the bounding box if the Intersection over the ground truth box exceeds 0.5, which is consistent with Section 3.1. Based on this, we measured the Accuracy, False Positive Rate (FPR), and False Negative Rate (FNR).
>
> Furthermore, to comprehensively assess the prompt sensitivity of this "object exists" check, we evaluated the verification performance across three different prompt variations:
>
> - P1 (Ours): Does the {target} exist in this image? Answer Yes or No.
> - P2: Does this image contain the {target}? Answer Yes or No.
> - P3: Is the {target} present in this cropped image region? Answer Yes or No.
>
> | prompt|verify_acc(%)|verify_fpr(%)|verify_fnr(%)|
> |---|---|---|----|
> | P1| **95.8**|10.7| **1.0** |
> | P2| 95.5| **9.5** |2.9|
> | P3| 95.6 |10.7|1.1|
>
> The results show that the verification module maintains high accuracy and a consistently low False Negative Rate (FNR) across different prompts, indicating strong prompt robustness and highly reliable verification.
>
> **W2: Implementation details for box-to-token mask injection**
>
> We describe the box-to-token mask injection step-by-step when a localized bounding box is verified as empty:
>
> 1. Visual Tokens: We identify visual tokens whose corresponding patch centers fall within the rejected box.
> 2. Text Tokens: We tokenize the target object description to locate its exact indices in the tokenized input query.
> 3. Masking: During the subsequent LLM inference to re-localize, we set the attention mask values from these specific text tokens to the identified visual tokens to $-\infty$.
>
> This efficiently suppresses the model's attention to distractors. The corresponding pseudocode is in Appendix A, lines 4-30. We will add this step-by-step explanation to the revision.
>
> **Q3: Missing Anchors**
>
> We counted the extracted semantic anchors on the $V^*$ Bench and found that 98.8% of the annotated objects were successfully recognized. Furthermore, even for the few samples with missing anchors, the model still answered correctly. This demonstrates that missing anchors are negligible exceptions, ensuring the robustness of our SAL module.
>
> **Q4: Sensitivity to the chosen attention layer and code release**
>
> We evaluated the sensitivity with respect to layers on Qwen3VL 3B and InternVL2.5 4B. As demonstrated in the table below, while peak performance is achieved at a specific optimal layer(layer 22 for Qwen3VL and 27 for InternVL2.5), ActiveScope remains highly robust across a range of adjacent layers. Notably, even at layers with sub-optimal performance, our method still yields substantial improvements over both the regular baseline models and the compared method ViCrop.
>
> | Model | Method | Layer| V*-Attr | V*-Spat| V*-Avg|
> |----|----|----|----|----|----|
> | Qwen3VL 3B|Regular| - | 86.96 | 93.42| 89.53|
> | | ViCrop |22| 86.96| 92.11| 89.01|
> |  | ActiveScope |21| **96.52** | 93.43| 95.29|
> |  | ActiveScope| 22| 95.65| **97.37** | **96.34** |
> |  | ActiveScope | 23| **96.52** | 94.74| 95.81|
> |  |ActiveScope| 24| 93.91| 93.43| 93.72|
> |  InternVL2.5 4B | Regular | - | 68.70 | 68.42| 68.59 |
> | | ViCrop|27|44.00|58.25|51.13|
> | | ActiveScope | 26| 74.80 | **82.89** | 78.01 |
> | | ActiveScope| 27| **80.00** | 81.58 | **80.63** |
> | | ActiveScope| 28 | 76.52 | 81.58 | 78.53 |
> | |ActiveScope | 29 | 78.26 |80.26 | 79.06 |
>
> We will release the source code, prompts, and evaluation scripts to ensure complete reproducibility.
>
> **L: societal impact not discussed**
>
> We would like to politely clarify that the impact statement is included on page 9, line 440 of the manuscript.

---

> > ### Author Rebuttal · Reviewer_xawK · 2026-04-03
> >
> > Thanks for the clarification and the additional results. I have read the rebuttal carefully.

---

> > > ### Author Response · Authors · 2026-04-07
> > >
> > > Thank you for reading our response carefully and confirming that your concerns are fully resolved. We would be very grateful if you could consider re-evaluating our work. Thank you again for your time and constructive feedback.

---

### Official Review · Reviewer_n1Py · 2026-03-13

**Soundness:** 3
**Presentation:** 3
**Significance:** 2
**Originality:** 2
**Overall Recommendation:** 4
**Confidence:** 2

**Summary:**

This paper studies the fine-grained perception limitations of VLMs on high-resolution images. Although recent multimodal models have improved their support for high-resolution inputs through dynamic resolution, variable numbers of visual tokens, or tile-based encoding, they still struggle to reliably identify critical details in complex high-resolution scenes. The paper argues that existing training-free “locate-then-zoom” methods remain fragile in such settings, with two main failure modes. The first is inaccurate localization, where the model is distracted by salient but irrelevant contextual regions. The second is incomplete localization, where in multi-object questions the model tends to attend only to the most salient object while missing other equally relevant ones. The authors attribute these two issues to Contextual Dominance and Semantic Bias, respectively.

To address this, the paper proposes ActiveScope, a training-free framework built on top of existing tile-based or dynamic-resolution input mechanisms. The method first leverages the model’s internal attention signals to construct finer-grained heat maps guided by question-relevant semantic targets. For models that process high-resolution images via tiles, ActiveScope first extracts attention maps from individual tiles and then reconstructs them into an image-level attention map for downstream localization. It then uses Semantic Anchor Localization (SAL) to generate object-specific localization cues for each key target, followed by Interference-Suppressed Refinement (ISR) to verify and suppress failed regions so that the next inference round can shift attention to alternative candidate regions. Finally, the verified crop is combined with the original image to produce the answer. Experiments on V* Bench, HR-Bench 4K/8K, and MME-RealWorld-Lite show that the method is effective under the target setting.

**Compliance With Llm Reviewing Policy:**

Affirmed.

**Final Justification:**

After considering both the paper and the authors’ rebuttal, I am changing my recommendation to weak accept.

The paper is sound, clearly presented, and well motivated, with a practical training-free method that addresses an important limitation of high-resolution VLMs. My main concerns were about evaluation breadth, system overhead, and applicability beyond the base setting. The rebuttal addressed these points credibly by adding overhead analysis, broader experiments, and scaling results on larger and MoE models.

**Key Questions For Authors:**

1. Since both SAL and ISR require access to attention maps and mask intervention, can this method be approximately implemented on closed-source models or models without transparent attention interfaces?
2. How does the method scale as the number of relevant objects increases, especially in dense scenes or OCR-heavy images? Could the authors provide curves such as “number of targets vs. accuracy” or “number of targets vs. number of calls”?
3. Could the paper include a small quantitative analysis on the relationship between localization quality and final QA accuracy?
4. In the efficiency section, could the authors also report the extra overhead relative to the base model, such as additional per-sample latency, the estimated cost of verification, and the system-level cost introduced by the iterative refinement pipeline?

**Limitations:**

1. Limited applicability to non-open models. Since the method depends on internal attention access and intervention, its applicability to closed-source or API-only models is limited.
2. Unclear behavior on larger-scale and MoE models. The paper does not study whether the method continues to work similarly for much larger VLMs or for MoE-based architectures, where attention behavior and inference dynamics may differ substantially.
3. Potential impact on inference orchestration and system engineering. Because the method introduces multiple stages, including attention extraction, reconstruction, verification, suppression, and re-inference, it may significantly complicate inference orchestration and engine design compared with standard one-pass VLM inference.

**Strengths And Weaknesses:**

## Strengths
1. The problem analysis is clear and the method is well motivated. This work first identifies two empirical issues, inaccurate localization and incomplete localization, and then further analyzes their root causes as contextual dominance and semantic bias. This makes the method better grounded than a purely heuristic test-time improvement.
2. The method is training-free and practically appealing. ActiveScope does not require additional training, SFT, or RL. It enhances high-resolution understanding through attention heat map analysis, which makes it relatively easy to use as a plug-in on top of existing VLMs.
3. The empirical results are strong under the evaluated setting. The proposed method consistently outperforms prior training-free baselines across multiple benchmarks. The efficiency analysis is also promising: compared with ZoomEye, it significantly reduces the number of VQA calls, suggesting practical value in addition to accuracy gains.

## Weaknesses
1. ActiveScope requires cross-attention maps and direct attention-mask intervention. This creates substantially different inference-system requirements and may require model-weight-level access. The paper should provide a more thorough analysis of the resulting system overhead, including a quantitative way to estimate the extra cost, and discuss the implications for inference deployment when direct weight or internal-state access is required.
2. The evaluation lacks analysis for other common high resolution scenarios . The paper does not provide enough analysis of more challenging real-world scenarios, such as highly dense scenes, OCR-heavy images, or long-chain compositional visual reasoning. As a result, the practical boundary of the method remains somewhat unclear.
3. There is limited analysis and experiment results on larger models and model scaling. The paper does not study how the method behaves on larger-scale models such as 30B or 70B VLMs, nor does it analyze whether attention-map behavior changes with model scale. It is also unclear how the method would need to be adapted for current MoE-based multimodal models.

---

> ### Author Rebuttal · Authors · 2026-03-29
>
> We sincerely thank you for your constructive feedback. We will address your concerns below and will incorporate these clarifications into the revised manuscript:
>
> **W1, Q4 \& L3: System Overhead**
>
> While ActiveScope introduces a multi-stage pipeline, the extra overhead is highly manageable. For localization, extracting the required attention maps only involves computing the cross-attention between text and visual tokens up to the intermediate layer, avoiding the higher cost of full inference and generation. During verification, the input bounding box crops are significantly smaller than the original high-resolution image, keeping computational costs modest. As discussed in Section 5.5, we provide a more detailed quantitative analysis of the system overhead evaluated on the V* Bench below:
>
> |Method|Latency(per sample) |Total Latency|VQA calls|GPU Memory|
> | ---- | ---|--- | ---- | --- |
> | Regular|4.4s|14min|1|36GB |
> | ZoomEye|8.5s|27min|33|36GB |
> | Ours|7.5s| 24min|4.7|40GB|
>
> Compared to the base model, ActiveScope adds only ~3.1s latency and ~4GB GPU memory per sample. Notably, it achieves more substantial performance gains than ZoomEye while requiring fewer VQA calls and lower latency.
>
> **W2 \& Q2: Evaluation on Diverse Scenarios and Impact of Object Count**
>
> We have conducted additional evaluations on dense scenes (MME-RW-Lite Remote Sensing & Monitoring), OCR-heavy images (TextVQA, DocVQA), and visual reasoning tasks (GQA, VSR). ActiveScope demonstrates consistent performance advantages across these challenging domains.
>
> |Mehod| Model|TextVQA| DocVQA|GQA|VSR|MME-RW-Lite Mt|MME-RW-Lite RS|MME-RW-Lite Avg|
> |----|:-----:|------|:-----:|-----|-----|:-----:|----|----|
> | Regular|Qwen3VL 3B|85.56|93.44 | 72.34 |78.14 |32.29|42.67|46.74|
> | ZoomEye|Qwen3VL 3B|86.21|94.01|72.82|78.66|37.30|46.00|49.40|
> | Ours|Qwen3VL 3B|**86.77**|**95.12**|**73.01**|**79.00**|**38.32**|**47.67** |**50.02**|
>
> Furthermore, regarding analysis of the number of objects(Q2), we have plotted the requested curves ("number of targets vs. accuracy" and "number of targets vs. number of calls") in [Figure R1](https://anonymous.4open.science/r/ActiveScope-D8EB/target_count_curves.png). The curves demonstrate that as the number of target objects increases, our method consistently maintains its performance advantage.
>
> **W3 \& L2: Scaling to Larger and MoE Models**
>
> We have evaluated ActiveScope on a larger model (InternVL2.5 26B) and a Mixture-of-Experts model (Qwen3VL-A3B 30B). The results show that our approach maintains its superiority over both the base models and the comparative method ZoomEye, confirming its robust effectiveness across larger-scale models and other architectures.
>
> | Mehod|Model| V*-Attr | V*-Spat | V*-Avg | HR4K-FSP | HR4K-FCP  | HR4K-Avg  |  HR8K-FSP | HR8K-FCP  | HR8K-Avg  |
> |----|:----:|:----:|-----|----|:------:|-----|----|:----:|----|---|
> | Regular | InternVL2.5 26B | 73.91| 72.37| 73.30|**83.50** |60.75|72.13|73.25| 61.50| 67.38|
> | ZoomEye | InternVL2.5 26B|74.78 | 73.68 | 74.35| **83.50** |63.00|73.25|75.75|63.00| 69.38|
> | Ours| InternVL2.5 26B|**80.00** | **75.00** | **78.01** |81.50| **68.00** | **74.75** | **79.25**|**63.25**|**71.25** |
> | Regular | Qwen3VL-A3B 30B(MoE) |86.73|86.49| 86.63 | 88.25| 63.50|75.88| 86.75 | 59.25 | 73.00 |
> | ZoomEye | Qwen3VL-A3B 30B(MoE) |**90.43**|85.53|88.48|89.25 | 64.25| 76.75| 88.25 | 59.50 | 73.88 |
> | Ours|Qwen3VL-A3B 30B(MoE)|89.38| **90.54** | **89.84** | **90.00** | **65.25** | **77.50**|**89.75**|**60.25**|**75.00**|
>
> **Q1 \& L1: Applicability to Closed-Source Models**
>
> We appreciate your insightful suggestion regarding closed-source models. We approximately implemented ActiveScope via iterative Q&A. Specifically, we replace the inaccessible attention maps with explicit bounding box predictions from the model, and prompt it with historical erroneous bounding boxes to refine its localization. As shown below, this approximate implementation yields modest improvements on GPT-4o, validating the generalizability of our framework on closed-source models.
>
> |  |  |V*-Attr| V*-Spat| V*-Avg|
> |---|---|---|---|---|
> |Regular|GPT 4o| 67.83|63.16| 65.97|
> |ActiveScope|GPT 4o|**68.26**|**64.47**|**66.75**|
>
> We would like to clarify that our core value lies in investigating and utilizing the **inherent localization patterns** of MLLMs, rather than treating MLLMs as unexplainable black boxes. Similar to existing perception-enhancing methods (e.g., ViCrop, ICoT), ActiveScope leverages internal representations to serve as a widely applicable framework for open-source models.
>
> **Q3:  Localization vs. QA Accuracy**
>
> As shown in [Figure R2](https://anonymous.4open.science/r/ActiveScope-D8EB/localization_vs_qa.png), we provide line charts evaluated on the V* Bench that plot localization quality against both the maximum cycle count $C$ and the final QA accuracy. The results show that a higher maximum cycle count $C$ optimizes bounding box localization, which directly improves final QA accuracy.

---

> > ### Author Rebuttal · Reviewer_n1Py · 2026-04-04
> >
> > I think the authors resolved my questions very well. I will raise the score.

---

> > > ### Author Response · Authors · 2026-04-05
> > >
> > > We deeply appreciate your thorough review and constructive suggestions. We are glad to know that our responses have resolved your questions, and we sincerely thank you for your updated assessment.

---

### Decision · Program_Chairs · 2026-04-30

**Decision:**

Accept (regular)

**Comment:**

The paper received 3x borderline accepts and 1x accept. It proposes ActiveScope, a training-free framework built on top of existing tile-based or dynamic-resolution input mechanisms. Reviewers n1Py, K3No, and MvN1 appreciate the insights of the paper and its comprehensive experimental validation. Reviewer xawK admires the motivation and the thorough ablation analysis of this work.
During the rebuttal, reviewers raised concerns regarding layer selection during reproduction, FP/FN rates, and prompt sensitivity. The authors addressed these concerns effectively by providing additional experiments. Reviewer K3No further questioned the deployment cost, and the authors responded with an analysis, showing that the lightweight process can accurately identify the optimal layers with very small computational overhead.
After considering the reviews and the rebuttal, the ACs recommend accepting this paper.